# Density Gradients, Cellular Structure and Thermal Conductivity of High-Density Polyethylene Foams by Different Amounts of Chemical Blowing Agent

**DOI:** 10.3390/polym14194082

**Published:** 2022-09-29

**Authors:** Juan Lobos, Saravana Prakash Thirumuruganandham, Miguel Angel Rodríguez-Pérez

**Affiliations:** 1School of Physical Sciences and Nanotechnology, Yachay Tech University, Urcuqui 100119, Ecuador; 2Centro de Investigación de Ciencias Humanas y de la Educación (CICHE), Facultad de Ingenieria y Tecnologias de la Informacion y la Comunicacion, Universidad Tecnologica Indoamerica, Ambato 180151, Ecuador; 3Cellular Materials Laboratory (CellMat), Condensed Matter Physics Department, Faculty of Science, Campus Miguel Delibes, University of Valladolid, Paseo de Belén 7, 47011 Valladolid, Spain

**Keywords:** improved compression moulding, polymer foams, LDPE, transient plane source method, cellular structure, thermal conductivity, density gradients

## Abstract

LDPE (low-density polyethylene) foams were prepared using the improved compression moulding technique (ICM) with relative densities ranging from 0.3 to 0.7 and with different levels of chemical blowing agents (from 1% to 20%). The density gradients, cellular structure and thermal conductivity of the foams were characterized. The density and amount of CBA used were found to have a significant effect on the cellular structure both at the mesoscale (density gradients) and at the microscale (different cell sizes and cell densities). In addition, the thermal conductivity of the samples is very sensitive to the local structure where the heat flux is located. The technique used to measure this property, the Transient Plane Source method (TPS), makes it possible to detect the presence of density gradients. A simple method for determining these gradients based on thermal conductivity data was developed.

## 1. Introduction

Polymer foams can be defined as two-phase materials in which a gas is dispersed in a continuous macromolecular phase [1]. Polymer foams are used in various fields due to their lightness, low thermal conductivity, high energy absorption and excellent strength-to-weight ratio. The uses of polymer foams are very diverse, including transportation, bedding, floor covering, fabrics, sports tools, insulating devices, construction, biomedical and automotive. Polymer foams can be flexible or rigid based on their cell geometry [2,3]. Due to their outstanding properties, thermoplastic foams have become indispensable items, and although they are mainly used as thermal insulators or shock-absorbing elements, they have found applications in almost all fields [3,4,5,6]. Since their development in the 1940s, both academia and industry have focused their attention on developing foaming technologies that can satisfy the growing demand for thermoplastic foam products [4]. Nowadays, no single foaming method dominates the production of thermoplastic foam, and both continuous and batch processes using chemical or physical blowing agents are operated. The most common foaming processes for thermoplastic foam production are extrusion, injection moulding, compression moulding, or batch foaming (dissolution of a physical blowing agent at high temperature and/or pressure) [1,6,7]. The choice of foaming process depends mainly on the final application of the material, which at this stage depends strongly on its relative density (i.e., the density of the foam divided by that of the corresponding solid). It is well known that both the foaming process itself and the intrinsic properties of the polymer matrix strongly influence the cellular structure of the foamed product [5,6,7].

We also note that previous studies of the thermal conductivity and relative density of various types of polymer foams generally yielded high values for relative density. For example, a study evaluating the thermal properties of various extruded and cross-linked polyethylene samples yielded a relative density value of 0.16 to 0.02 [8], as did polystyrene/nanocomposite foams made from multi-walled carbon nanotubes for thermal insulation applications, ensuring that all foams in that study had a relative density value of 0.125 to 0.025 [9]. In addition, a study of ultra-high expansion linear polypropylene foams synthesized in the semi-molten state under supercritical CO_2_ conditions confirmed that all foam samples had a relative density of 0.05 to 0.02 [10]. Recently, thermal conductivity measurements confirmed a high density value of 0.1 [100 kg/m^3^] for LDPE through detailed investigations using a response surface methodology [11]. In conjunction with previous findings, our goal is to investigate the final density, cell structure and coalescence. In addition, we should examine whether the nucleation was homogeneous or heterogeneous with the rest of the azodicarbonamide. For this reason, we prepared a sample with 20% azodicarbonamide and twice the amount of residue to duplicate the nucleation points that we summarize in the Methods Section in (Table 1) to vary the nucleation points and coalescence.

Several years ago, the CellMat laboratory developed a new method for the production of foams, called improved compression moulding (ICM) [12,13,14,15]. In this method, the foam density is controlled by using special molds called self-expandable molds. These molds are capable of exerting and maintaining pressure during decomposition of the chemical blowing agent, subsequent dissolution of the gas phase and foaming. In addition, the molds are able to control the degree of expansion of the material, thereby regulating both the size and shape of the foamed part very precisely. The internal pressure of these molds is determined by the amount of blowing agent in the precursor. The advantage of this technique is that the polymer does not require a crosslinking additive [16,17], so the defective parts can be reused in the process. An interesting aspect of this technique is that it allows the production of foams with the same density but very different cell structures, by changing the chemical composition and/or the foaming parameters. Therefore, it is possible to produce interesting samples for scientific studies. In the present work, foamed polyethylene samples are prepared with different amounts of blowing agent (azodicarbonamide, from 1% to 20%), similar to [12,18]. This allows us to obtain a collection of samples with different cell structures, both at the mesoscopic and microscopic level. The prepared samples were characterized by their relative density, thermal conductivity, cell structure and X-ray images. The main objective of this work is to gain knowledge about the structure–property relationships of these materials, in particular, the dependence between the structure and the thermal conductivity of these materials.

## 2. Materials and Methods

The raw materials used to produce the LDPE foams were:Low-density polyethylene (PE008) from Repsol, Quimica, Tarragona, Spain with a melting temperature of 110 °C, MFI = 4 g/10 min at 190 °C, density 923 Kg/m^3^.Azodicarbonamide (AZO) from Uquinsa Barcelona, Spain as the blowing agent, with an average particle size of 4.9 µm.Zinc oxide, Silox Active grade from Safic Alcan, Barcelona, Spain as the catalyst for the chemical blowing agent.Stearic acid (Stearic Acid 301), supplied by Renichen, Barcelona, Spain is used as a processing aid.

We note that the polymer is the basic component of the material in which the azodicarbonamide (AZO) was used as a blowing agent to decompose it by temperature and to release the gas and form the seed particles in the plastic. The zinc oxide reduces the decomposition temperature of the azodicarbonamide from 210 °C to only 185 °C. In addition, the stearic acid is used as a lubricant to mix everything in the Rheodrive 5000 with the Rheomix 500 mixing head at 145 °C, 25 rpm, for 5 min, with a total mass of 50 g. The mixture was crushed into pieces of 2–8 mm^3^ volume (pellets). Homogeneity was checked by thermogravimetry experiments on three samples of the mixture.

In our technique, the scheme for the preparation of the precursor was adopted from the already established processing technique in European Patent No. WO2008081054A1 (2008-07-10) [13]. The size distribution of the bubbles is an important parameter on the moulded part, which depends on the characteristics of the blowing process. All bubbles form at once, due to the adiabatic decompression, to create a foam, so we can control the volume of a foam in advance. On the other hand, a desired volumetric fraction of a foam can be decided in advance. The proposed method could also be used to study the arrangement of bubble deposits of different sizes, which is directly related to the mechanical properties of the workpiece. For all these objectives, it is necessary to make a precursor, as shown in Figure 1, and the A to D sub-figures show the processing steps.

Furthermore, the foam is made as is shown in Figure 2, to have control over the morphology and final volume.

Thermal decomposition of azodicarbonamide occurs at 210 °C without a catalyst and at 185 °C with 0.05% ZnO [19,20]. During decomposition, azodicarbonamide releases 25.4% of its weight as gases at 190 °C (N_2_ 71.9%, CO 26.3%, CO_2_ 1.4%, NH_3_ 0.4%), leaving 74.6% solid residues in the polymer [20,21]. For the production of high-density foams, it is sufficient to add 1–2 wt% of azodicarbonamide into the polymer [14,22,23]. However, due to the special design of the molds used in this work, much higher amounts can be used to modify the cell structure. Table 1 shows the amounts of azodicarbonamide chosen (from 1 to 20%). Afro-c + R is a special composition with 10.5% azodicarbonamide and 6.6% azodicarbonamide residues. The azodicarbonamide residues were generated by heating the blowing agent to 190 °C for 10 min. This material had the same number of solid residues as Afro-e, but a very different pressure during the foaming process. The Improved Compression Moulding (ICM) route involves the following steps. A certain number of pellets are placed in the mold (23 mm diameter, 27.5 mm height) and subjected to both temperature (TP = 120 °C) and pressure (PP = 5 Tn) to ensure good compression of the pellets, with the aim of producing the precursor material (Figure 1B). The density of the precursors ranged from 925 kg/m^3^ to 957 kg/m^3^, depending on the amount of azodicarbonamide in the composition. The pictures of the experimental setup of the mold and the foam developed in the laboratory can be found in Figure 1, Figure 2 and Figure 3.

In the second and final step (Figure 2A–D), the precursor is introduced into the foaming mold (23 mm diameter and 27 mm height), which is located in a foaming press. An initial pressure (P0 = 15 Tn) is applied while the mold is heated up to the foaming temperature (TF = 190 °C) (Figure 2 A–C). As the temperature increases, the blowing agent begins to decompose and the pressure in the mold increases to a higher value (PF). After a certain time (tF = 900 s), when the blowing agent has completely decomposed and PF stabilizes (does not increase further), the pressure of the press is released, allowing the polymer to expand until a desired volume is reached (Figure 2D). The mold is cooled with water and air for 5 min to reach room temperature (Tr = 26 °C). The finished foam had a height of 23 to 25 mm and a diameter of 22.5 mm (Figure 1D). Density characterization was performed using the hydrostatic weighing set of the Mettler Toledo AT261 balance. Relative density (density of the foam divided by the density of a solid precursor) is used in this work to make comparisons between materials. X-rays were used to analyze possible density gradients in the materials. After the density was measured, the top of the sample was cut to the same height, 18 mm, for all samples (Figure 4b. The upper part (7 mm thick) was used to characterize the cellular structure by SEM microscopic images using a JEOL JSM −820 microscope (Peabody, MA, USA) after the samples, previously cooled in liquid nitrogen, were broken. The SEM images were analyzed using ImageJ software to obtain the cellular parameters of the foam. The lower part of the samples (18 mm thick) was used to characterize their thermal properties using the Transient Plane Source (TPS) method.

The TPS method uses a round and flat heat source. It behaves like a transient flat source that also works as a temperature sensor. This element consists of an electrically conductive pattern of thin nickel foil (10 µm thick) in the form of a double spiral sandwiched between two insulating layers of Kapton (70 µm thick), so that the final sensor thickness is 150 µm. The element TPS is located between two samples, with both sensors in contact with similar density and the “hot disc” sensor. The temperature rise ∆*T*(*t*) is directly related to the change in sensor resistance *R*(*t*) by equation [24]:(1)$$R\left(t\right)=R_{0}\left(1+\alpha \Delta T\left(t\right)\right)$$
where *R*_0_ is the disc resistance at the beginning of the recording (initial resistance) and *α* is the temperature coefficient of resistance of the nickel foil. Assuming an infinite sample and that the conducting pattern lies in the XY plane of a coordinate system, the temperature rise at a point (XY) at time t is obtained by solving the equation for heat conduction relating the temperature change to time [25,26,27]. In the case of our sensor geometry, n concentric ring sources, the spatial average can be obtained by the equation:(2)ΔT(τ)=P0(π32αλ)−1D(τ)
where *P*_0_ is a Bessel function, *D*(*τ*) is a geometric function characteristic of the number n of concentric rings and ∆*T*(*τ*) is the temperature rise of the sensor expressed by only one variable *τ,* defined as:(3)τ=(tθ)12; 
(4)$$\theta =\frac{a^{2}}{k}$$
where *t* is the measurement time from the onset of transient heating, *θ* is the characteristic time that depends on the parameters of the sensor and sample, a is the radius of the sensor and *k* is the thermal diffusivity of the sample. The thermal conductivity can be obtained by fitting the experimental data to the straight line given by Equation (1), and the thermal diffusivity is calculated from Equation (3) by considering the experimentally determined *θ*-value. Thermal property measurements were performed on two samples with the same relative density, as shown in Figure 5. The samples were characterized in two different regions, the inner part (Figure 5a) and the outer bottom region (Figure 5b).

A sensor with a radius of 3.189 mm was chosen. Samples were shipped to ensure that the boundary conditions for performing the experiments with the transient plane radiation source were verified (Equations (2) and (3)). The selected output power (W) was 0.02 W and the total measurement time (*t*) was 20 s for all samples studied. At least five measurements were performed for each configuration. The probing depth ranged from 3.5 to 4.2 mm in the core and from 3.2 to 3.8 mm in the skin. In analyzing the density of foams, we use “relative” units. 1 is the density of LDPE without pores, and the relative density of 0.5 is half the polymer and half the volume occupied by air. The thermal conductivity of the foams is also relative, divided by the thermal conductivity of the LDPE to directly use the relative property, as described in Gibson and Ashby’s book *Cellular Solids: Structure and Properties* [3].

## 3. Results

### 3.1. Density Gradients

Figure 6 shows the X-ray image of one of the specimens (relative density) and the density as a function of position (distance from the skin in the direction of the cylinder diameter) for this particular material. The density was calculated from the X-ray image [28]. From the analysis, it appears that there is a significant density gradient. The density near the surface is 600 kg/m^3^ and in the core it is about 375 kg/m^3^. The density reaches a constant value at distances around 7 mm. This type of density gradient was observed both in the thickness direction and in the diameter direction and for all the samples studied.

The same effect can be observed at the micro level (Figure 7). Figure 7 shows a microscopic image of a foam with a relative density of 0.5. The cell size is significantly smaller in the area close to the skin, indicating a higher density in this area. It is also clear that the cell structure changes progressively from the skin to the core.

### 3.2. Cellular Structure

Considering the previous density and cell structure gradients, the comparative study of the cell structure of foams prepared with different amounts of blowing agents and bulk densities was carried out for samples cut from the inner part of the foam. The cell structure changes significantly when the density is changed. Figure 8 and Table 2 show the images of SEM and the cellular properties of the foam center for three objective densities when the same amount of blowing agent is used (10.5%).

As expected, foam cell size decreases as foam porosity is reduced (higher density); images from SEM show a significant degree of cell coalescence at low densities. Assuming that the CBA residues act as nucleation sites for the cells, it is possible to calculate the density of the nucleation sites (N-P/cm^3^ in Table 2) [28,29,30]. Based on this value divided by the cell density, we determined the number of nucleation sites required to form a single cell in the final foam. For the high-density foams, coalescence is low, only 5.4 nucleation sites are needed to generate a cell, and there is almost no coalescence. This value increases to 20.1 for the medium-density foams and is about 30 times higher for the low-density foams. On the other hand, Figure 9 and Table 3 show the effects of CBA content on the cellular structure of foams with the same relative density, 0.5.

The results show that increasing the CBA content up to 10% allows a significant reduction in cell size and an increase in cell density. The cell size decreases from 100 μm for the material prepared with 1.2% CBA to about 40 μm for the material with 10% CBA content. For the materials containing 15 and 20% azodicarbonamide, the cell size increases slightly. Interestingly, the material prepared with 10% active CBA and 10% CBA residues shows a smaller cell size and a higher cell density. The obtained results can be explained considering several simultaneous effects. On the one hand, it is clear that the propellant residues serve as nucleating sites for the cells. On the other hand, increasing the CBA content up to 10% seems to have a positive effect on the stability of the foam, since the number of nucleation sites for the formation of a cell (N-P) is reduced at this CBA content. This effect worsens at higher pressures.

### 3.3. Thermal Conductivity

Figure 10 shows the experimental values of the thermal conductivity of the foams as function of the density, both of them relatives (i.e., the real value was divided by the value of the solid material). Experimental values for the solid PE were 923 Kg/m^3^ and 0.378 W/m·K.

For each density, there are two data in the figure. The data for the ‘core’ and the ‘skin’ are plotted as described in Figure 3. The average of these two data is also included in Figure 8. Ashby introduced a model to study the evolution of any physical property of a foamed material with density. This equation for thermal conductivity is as follows [3]:(5)$$\frac{\lambda }{\lambda_{s}}=\xi_{1}{\left(\frac{\rho }{\rho_{s}}\right)}^{n}$$
where lambda (*λ*) is the thermal conductivity of the foam, *λ_s_* is the thermal conductivity of the solid material, the density of the material is *ρ* and the density of the polymer is *ρ_s_*, 923 Kg/m^3^. The constants *ξ_1_* and *n* depend on the microstructure of the foamed material. The previous equation must be modified when the gas phase cannot be neglected, as in the case of polymer foams. To include the thermal conductivity of the gas phase, Equation (5) is modified as follows.
(6)$$\frac{\lambda }{\lambda_{s}}=\xi_{1}{\left(\frac{\rho }{\rho_{s}}\right)}^{n}+\xi_{2}{\left(1-\frac{\rho }{\rho_{s}}\right)}^{n}$$

This equation has a new constant *ξ*_2_ related to the contribution of the thermal conductivity of the gas phase. The experimental data were fitted to Equation (6). The best fit was obtained for *n* = 1.6 ± 2, *ξ*_1_ = 0.99 ± 0.05 and *ξ*_2_ = 0.15 ± 0.08 and R^2^ = 0.9016. The relative conductivity of the polymer is 1 (0.378 W/m·K) and the relative conductivity of the gas (air) is 0.07 (0.026 W/m·K). The fit for Equation (6) reproduces very well the average data obtained for the ‘skin’ and ‘core’ of the foams. Considering that the probing depth of the hot-disc TPS measurements was less than the effective distance at which the density reaches a constant value, it is possible to use the fit of the data to calculate an average value of the density in the region where the thermal conductivity is measured. Figure 9 illustrates this idea with an example. For the foam with a bulk relative density of 0.53 (494 kg/m^3^), the thermal conductivity data for the skin region can be understood if we assume an average density of the skin of 0.636 (423 kg/m^3^), as is shown in Figure 11. Therefore, the density gradient for this particular material can be determined as the difference between the two previous values. This result is in agreement with the measurements of the X-ray images.

The previous procedure was used for the foams with a relative density of 0.5 and 0.2 produced with different amounts of CBA (Figure 12).

In general, it can be observed that the increase in density versus macroscopic bulk density in the skin is greater than the decrease observed for the density of the core. This is logical considering the smaller thickness of the skin area. For the intermediate-density samples (relative density of 0.5), the differences between the higher density in the skin and the lower density in the core are almost constant, with a slight decrease when the amount of azodicarbonamide is reduced. This means that the density gradient is almost constant. For the low-density samples (0.2 relative density), there is a clear dependence on the CBA content. The density gradient is higher for the foams prepared with higher amounts of azodicarbonamide. This effect can be confirmed by observing the cell structure of one of these materials (Figure 13). It can be observed that due to the foaming process generated by the gas, several cells are crushed near the skin, increasing the density in this area.

As described in the experimental procedure, the foams were produced in 1200 s, with 900 s for heating and 300 s for cooling. When the foam is cooled in the mold, the outer part solidifies first, and the core of the foam is still in a molten state with a large amount of gas inside. As a result, and for foams with a low density and high blowing agent content, the cells near the skin are deformed, leading to late coalescence of the cells [29,30,31] and increasing the density. Furthermore, the core has more time to cause coalescence of the cells and the cell diameter is larger. The high pressure inside the plastic leads to a higher amount of polymer along the walls of the mold. Thus, the model of this material is as follows (Figure 14), as predicted by [32], and the rapid cooling could be used in various production lines.

## 4. Conclusions

A series of LDPE foams with different densities were produced with different amounts of chemical blowing agent. Thanks to the special properties of the improved compression moulding technology, it was possible to produce foams with constant density and different cell structures. The cell structure at the mesoscopic and microscopic levels was evaluated. This showed that the foams have a skin–core morphology and the residues of azodicarbonamide have a strong effect as nucleating sites for the cells. It has been shown that there is an optimal amount of blowing agent, 10%, when the goal is to reduce cell size. At this specific level, the strong nucleation promoted by the blowing agent residues is combined with stabilization of the cell structure by the pressure exerted on the mold. The characterization of the local thermal conductivity using the TPS method made it possible to determine the relationship between density and thermal conductivity for these materials and, taking this dependence into account, to determine the global density gradient that occurs during processing. It has been shown that the density gradients are stronger when foams with a low density are produced and when higher amounts of CBA are used. The azodicarbonamide foams are generated by heterogeneous nucleation, as the higher number of nucleation points is the sample with 10.5% azodicarbonamide, but double the amount of azodicarbonamide residues with the higher number of cell densities, as shown in Table 3. The core–skin morphology is due to the collapse process of the samples in the steel mold. Furthermore, it can be used in another process to produce polymer foams.

## Figures and Tables

**Figure 1 polymers-14-04082-f001:**
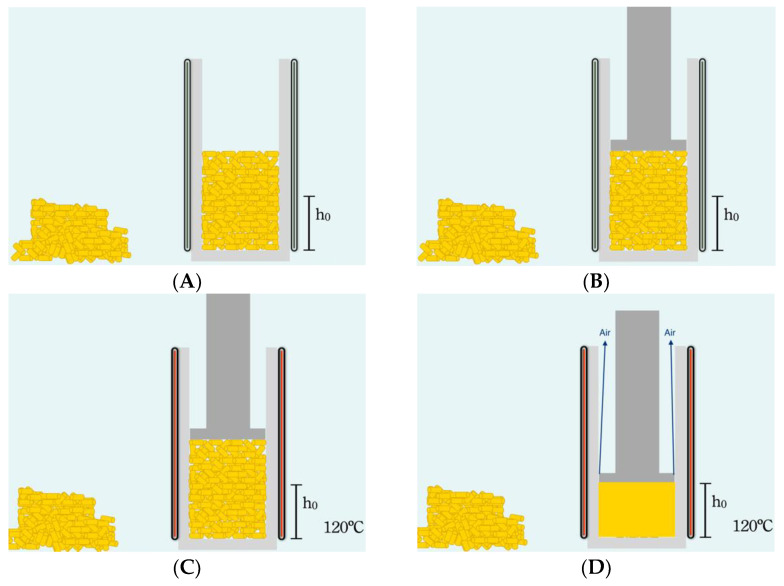
Schematics of the fabrication of the foam precursor. (**A**) Insert the pellets into the mold, (**B**) Close it with a piston, because the piston is loose, and the air could escape. (**C**) Heat at 120 °C to melt the polymer at 5 Tn and obtain a cylinder without trapped air. (**D**) We check all precursors by mass and density to be sure there is no air trapped in them. The density of the precursors ranged from 925 kg/m^3^ to 957 kg/m^3^, depending on the amount of azodicarbonamide in the composition.

**Figure 2 polymers-14-04082-f002:**
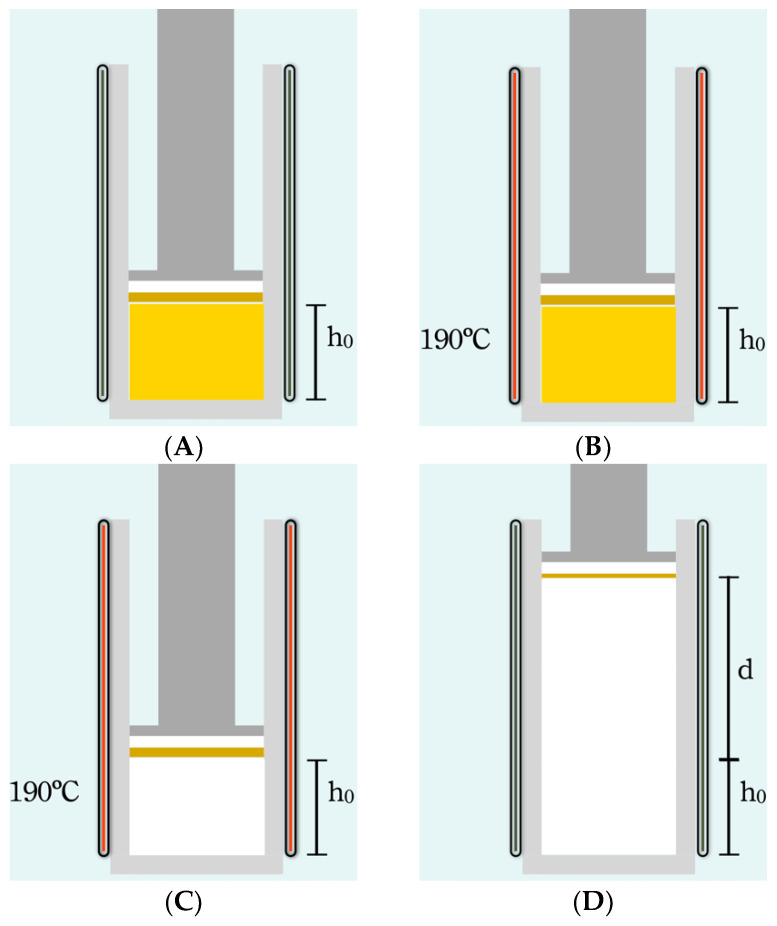
Making the foam from the precursor. (**A**) The foam was made in the same mold; we use a tight piston that does not allow the gas to escape. We seal it with 5 Tn. (**B**) Heat it to 190 °C to decompose the AZO. The hydraulic press increases the load when the gas is released, up to 18 Tn for the lower-density foams 250 Kg/m^3^, up to 30 Tn for the higher-density foams 750 Kg/m^3^. For this reason, we could not make all the compositions for the higher-density foam, but only the 750 b (5.8% AZO) and 750 c (10.5% AZO). The other compositions break the seal. (**C**) When the pressure reaches the maximum, the entire AZO is decomposed; we turn off the heater and let the piston go up. (**D**) The increase in volume forms all the bubbles in the polymer. We cool the steel mold with water spray and squeeze the air for 5 min. Some of the foams broke when we removed the foam from the mold because there were still large amounts of gas in the core of the foam.

**Figure 3 polymers-14-04082-f003:**
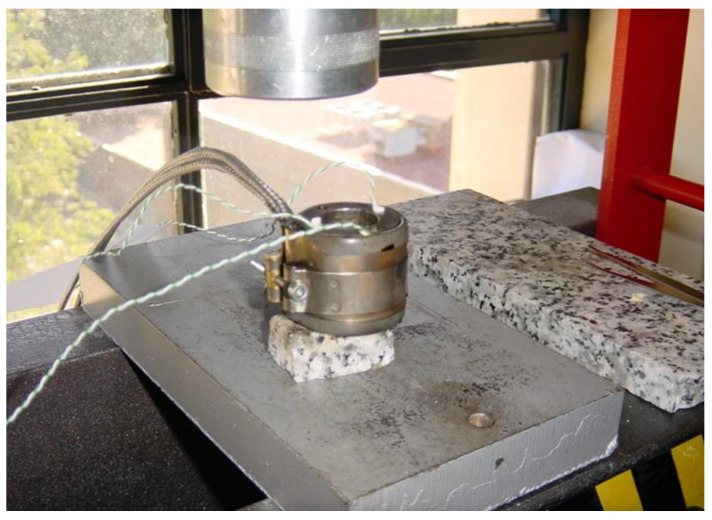
Experimental setup of mold with all the thermocouples in the mold to control the temperature.

**Figure 4 polymers-14-04082-f004:**
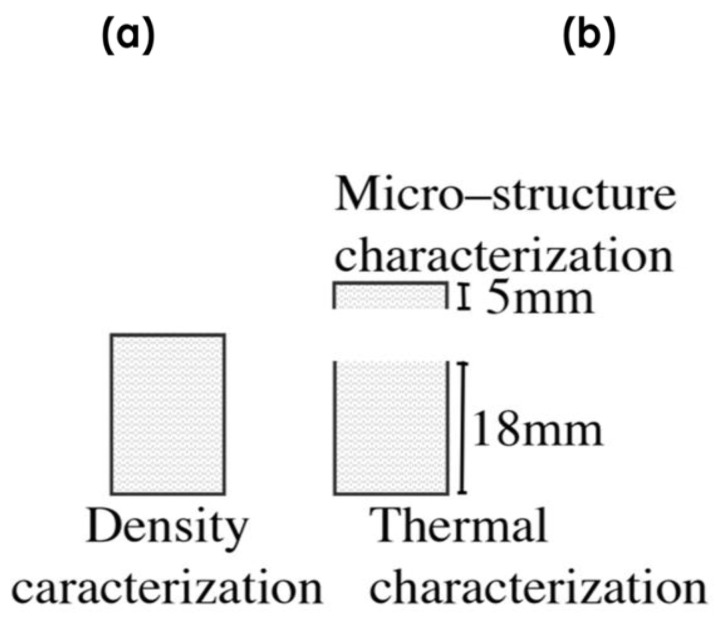
(**a**) Sample used to measure density. (**b**) Samples are used to characterize the cellular structure and thermal conductivity.

**Figure 5 polymers-14-04082-f005:**
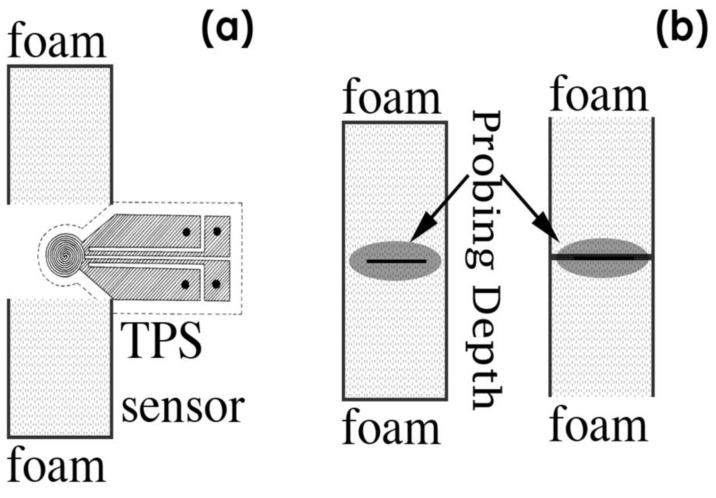
Measurement of the thermal properties of the foams by the TPS method. The TPS sensor is placed between two foams of the same relative density and produced using the same pellets. The sensor heats the foams by a ‘power output’ inside the volume delimited by the ‘probing depth’. For each couple of samples, the experiments were done in two different configurations. (**a**) Measurement in the foam core. (**b**) Measurement performed in the outer bottom surface (skin).

**Figure 6 polymers-14-04082-f006:**
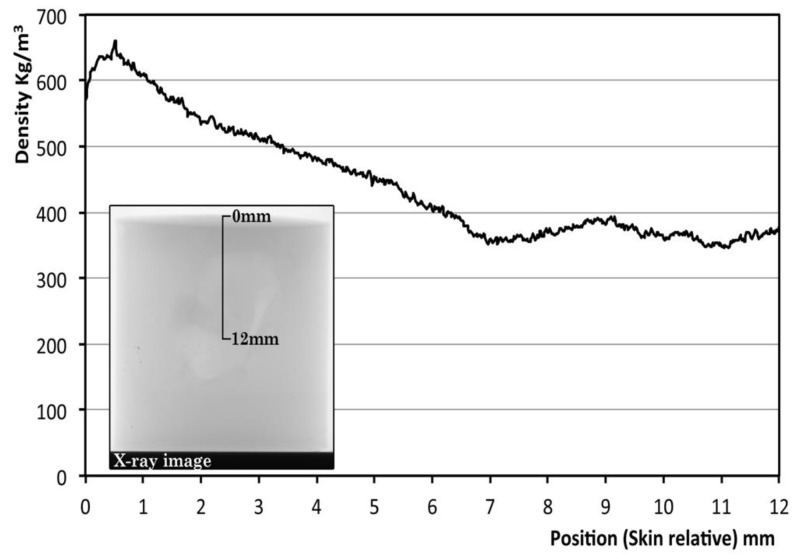
Internal density of one foam of 0.5 relative density, from the top to the center of the foam in the direction of the axis of the cylinder. The density is obtained by the intensity of grey in the X-ray image.

**Figure 7 polymers-14-04082-f007:**
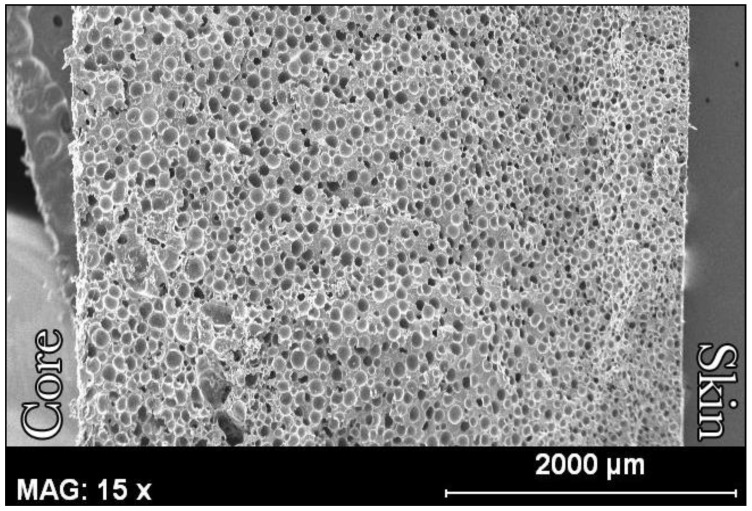
Fracture of the upper part of a PE foam with a relative density of 0.5. The axis of the cylinder is between the words ‘Skin’ (surface of the foam) and ‘Core’ (internal part of the foam).

**Figure 8 polymers-14-04082-f008:**
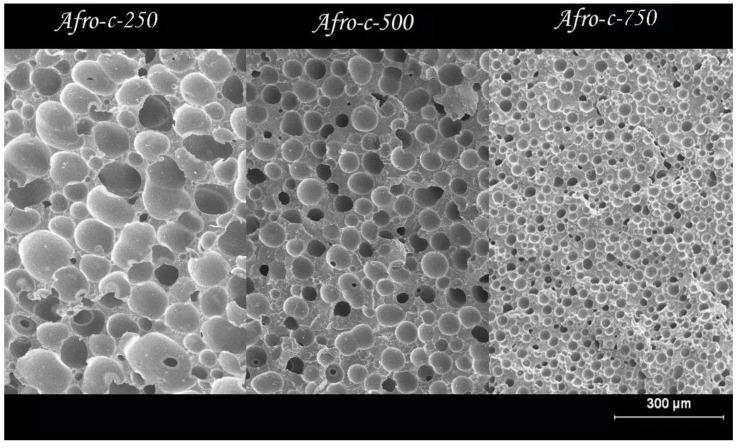
SEM images of the cellular ρ of foams produced with the same amount of blowing agent and three different relative densities.

**Figure 9 polymers-14-04082-f009:**
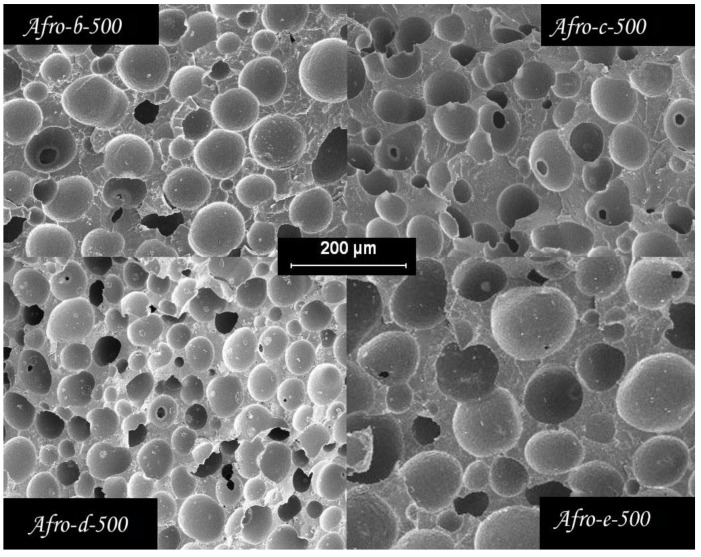
SEM images of the core of four foam samples with relative densities of 0.5 and different amounts of CBA.

**Figure 10 polymers-14-04082-f010:**
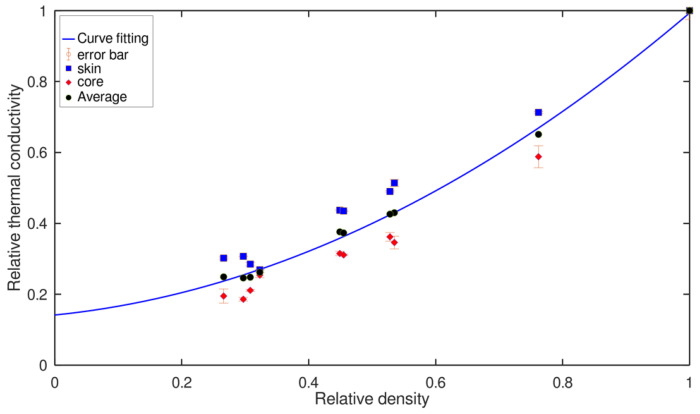
Relative thermal conductivity of the foams under study as a function of the relative density. The curve fit is Equation (6), where *ξ*_1_ = 0.99 ± 0.05 is the contribution of the polymer and *ξ*_2_ = 0.15 ± 0.08 is the contribution of the gas phase, the power is *n* = 1.6 ± 2 and the R^2^ = 0.9016.

**Figure 11 polymers-14-04082-f011:**
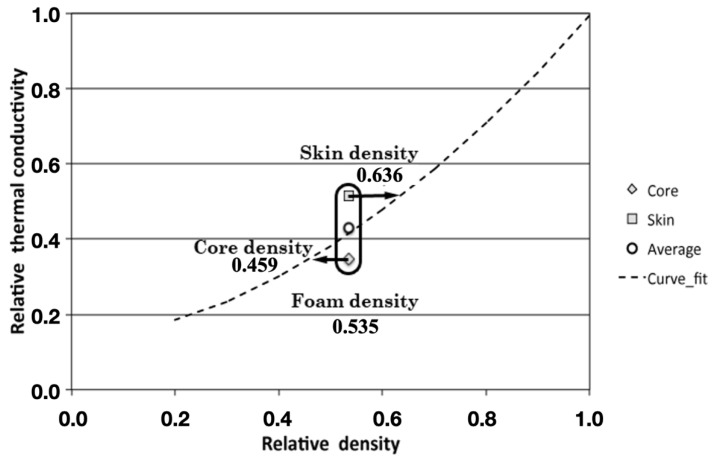
Different densities on the foams calculated using the thermal conductivity results.

**Figure 12 polymers-14-04082-f012:**
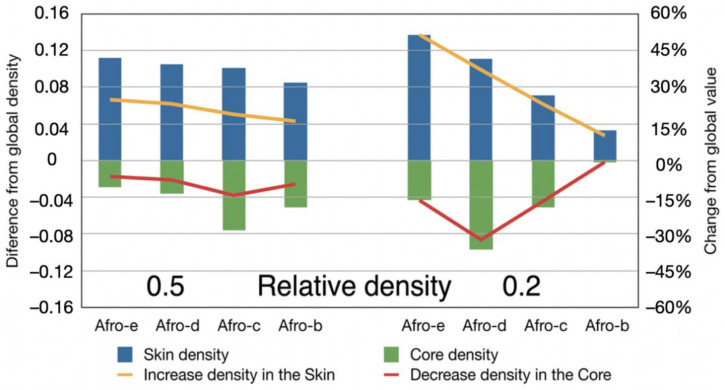
The yellow line shows the increase in density in the ‘skin’, and the red line shows the decrease of density in the ‘core’.

**Figure 13 polymers-14-04082-f013:**
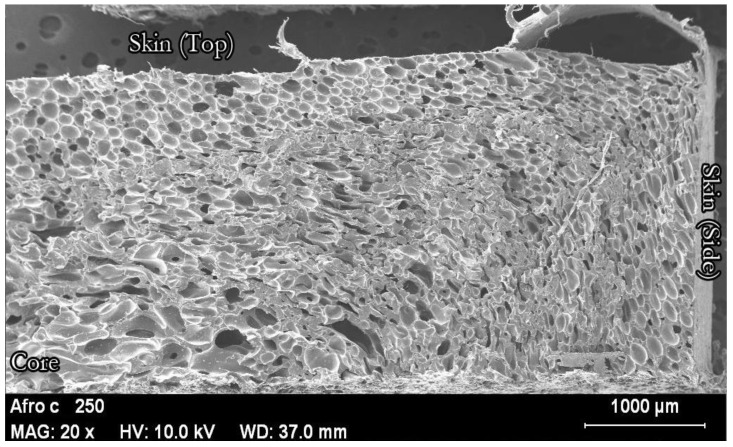
Sample with a relative density of 0.2 and produced with 10.5% of azodicarbonamide. Near the skin the cellular structure shows a regular structure. 500µm from the skin, several cells are crushed in the direction from the core to the skin. In the core, the cells are four times larger than in the skin.

**Figure 14 polymers-14-04082-f014:**
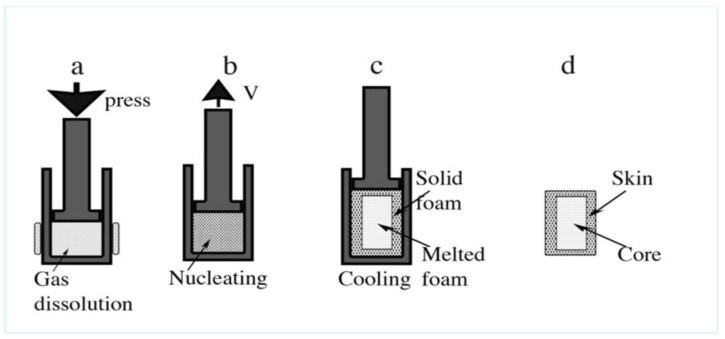
Schematic representation of the production of the foams. (**a**) melting of the polymer and decomposition of the azodicarbonamide, when the pressure stops increasing, the flask is released. (**b**) to carry out an adiabatic expansion and generate all the bubbles at the same time and fill the new volume. (**c**) the flask is stopped, and the mold is cooled down after a certain time, but due to the thermal conductivity of the porous material, the inner part still melts and allows higher coalescence in the core of the foams. (**d**) the final cylindrical sample had two different parts, but without a hard change, is gradually for the small pores in the skin to become a larger in the core of the samples.

**Table 1 polymers-14-04082-t001:** Composition of the foam precursors and the final residues of azodicarbonamide disperser in the final foams.

	Afro-A	Afro-b	Afro-c	Afro-d	Afro-e	Afroc + R
PE	98.6%	94.4%	89.3%	84.2%	79.3%	82.5%
AZO	1.2%	5.8%	10.5%	15.6%	20.5%	10.5%
AZO residue	-	-	-	-	-	6.6%
Octadecanoic acid	0.15%	0.15%	0.15%	0.15%	0.15%	0.15%
ZnO	0.05%	0.05%	0.05%	0.05%	0.05%	0.05%
Residues of azodicarbonamide in the polymer in the final foam						
	Afro-A	Afro-b	Afro-c	Afro-d	Afro-e	Afroc + R
	0.8%	3.7%	6.8%	10.1%	13.2%	13.4%

**Table 2 polymers-14-04082-t002:** Relative density, cellular diameter, cell density of foams, nucleating sites (N-P/cm^3^) and number of nucleating sites needed to produce a cell (N-P/cell) produced with a CBA content of 10.5% and different relative densities.

Afro-c	CBA Content	ρ (Relative)	Cell Size (µm)	Cell Density/cm^3^	N-P/cm^3^	N-P/Cell
(250)	10.5%	0.29	88.77	6.70 × 10^+06^	1.01 × 10^+09^	150.1
(500)	10.5%	0.51	38.60	4.91 × 10^+07^	1.01 × 10^+09^	20.1
(750)	10.5%	0.73	21.94	1.86 × 10^+08^	1.01 × 10^+09^	5.4

**Table 3 polymers-14-04082-t003:** Relative density, cell diameter, cell density, nucleating sites (N-P/cm^3^) and number of nucleating sites needed to produce a cell (N-P/cell) for the foams of medium relative density 0.5.

(500)	CBA Content	ρ (Relative)	Cell Size (µm)	Cell Density/cm^3^	N-P/cm^3^	N-P/Cell
Afro-a	1.2%	0.61	100.9	5.98 × 10^+05^	1.15 × 10^+08^	192.2
Afro-b	5.8%	0.53	79.4	5.43 × 10^+06^	5.56 × 10^+09^	102.4
Afro-c	10.5%	0.51	38.6	4.91 × 10^+07^	1.01 × 10^+09^	20.1
Afro-d	15.6%	0.46	45.5	3.32 × 10^+07^	1.49 × 10^+09^	45.0
Afro-e	20.5%	0.45	64.1	1.21 × 10^+07^	1.96 × 10^+09^	162.0
Afro-c + R	2 × 10.5%	0.56	29.0	1.05 × 10^+08^	1.72 × 10^+09^	19.1

## Data Availability

Not applicable.

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
