# Peer review of "Density Gradients, Cellular Structure and Thermal Conductivity of High-Density Polyethylene Foams by Different Amounts of Chemical Blowing Agent"

_polymers, 2022, doi:10.3390/polym14194082_

Round 1

Reviewer 1 Report

The manuscript "Density gradients, Cellular structure and thermal conductivity of high density polyethylene foams by different amounts of 3 chemical blowing agent" is well written and presented. However, it needs improvement before publication.

1. Introduction is not enough to judge the research gaps and problems. More literature need to be added. Compare the previous finding and extract the research gaps and frame objective of study.

2. Highlight the novelty and application area of research work.

3. Materials and Methods need to be improved. Describe the methodology in detail.

4. Add experimental set-up images and as-developed foam.

5.  The state-of-the-art comparisons for the proposed work are missing in this paper. Then do a critical analysis of previous research. State explicitly the shortcomings of previous research. What is positive in previous research and what is negative. Based on that, you explicitly define the goal of the research and the scientific hypothesis.

6. Highlight the novelty of your methodology.
7. The biggest shortcoming of the research is that there is no analysis of errors, analysis of sensitivity of results and analysis of uncertainty of results.
8. How did you choose the experiment setting? Elaborate experiments parameters.
p. The Conclusion section should be rewritten. Highlight your scientific contribution. Highlight the benefits of your research. Define shortcomings and future research.

Author Response

Dear Reviewer, Polymers-MDPI

ID# Polymers-1901227

On behalf of our research team, I thank you for considering our work.

In communication to our previously submitted manuscript with Polymers (21st of August, 2022)  and the Reviewer # 1 (29 Aug 2022 18:03:38) gave the following comments.

The manuscript "Density gradients, Cellular structure and thermal conductivity of high density polyethylene foams by different amounts of 3 chemical blowing agent" is well written and presented. However, it needs improvement before publication.

  1. Introduction is not enough to judge the research gaps and problems. More literature need to be added. Compare the previous finding and extract the research gaps and frame objective of study.
  2. Highlight the novelty and application area of research work.
  3. Materials and Methods need to be improved. Describe the methodology in detail.
  4. Add experimental set-up images and as-developed foam.
  5. The state-of-the-art comparisons for the proposed work are missing in this paper. Then do a critical analysis of previous research. State explicitly the shortcomings of previous research. What is positive in previous research and what is negative. Based on that, you explicitly define the goal of the research and the scientific hypothesis.
  6. Highlight the novelty of your methodology.

  7. The biggest shortcoming of the research is that there is no analysis of errors, analysis of sensitivity of results and analysis of uncertainty of results.

  8. How did you choose the experiment setting? Elaborate experiments parameters.

  9. The Conclusion section should be rewritten. Highlight your scientific contribution. Highlight the benefits of your research. Define shortcomings and future research.

I am attaching the reply for the review as a document.

Reviewer 2 Report

Equations must be rewritten to avoid overlpping.

In Fig. 8, show regression equation and R-square.

How was the thermal conductivity and relative conductivity measured/calculated? Give details in the methodology section.

The data is not analyzed statistically. Please add error bars and express R-square values.

Comma in figures should be replaced by decimal points.

The methodology section needs major revision and authors must add the detailed descriptions. There are several studies about thermal condustivity measurements. It should be referred to. Too much self citation observed.

Author Response

  Dear Reviewer, Polymers-MDPI

  ID# Polymers-1901227

On behalf of our research team, I thank you for considering our work.

In communication to our previously submitted manuscript with Polymers (21st of August, 2022) and the Reviewer # 2 (29 Aug 2022 18:03:38) gave the following comments.

Our comments are attached.

Reviewer 3 Report

Polymeric foams, due to their wide usage, receive high attention as research objects. Their properties are defined by the density, morphology and microstructure, thermal conductivity, stability, etc. Despite of quite an extensive research for several last decades, there are still lots of open questions relating, for instance, to the interrelation of production technology to particle size and morphology, material density and their impact on thermal properties and stability. In relation to this, the present work providing an insight into the density gradients, cellular structure and thermal conductivity of the polyethylene foams produced with improved compression moulding procedure and different content of blowing agent is actual. Using a set of morphology and thermal conductivity sensitive techniques, the authors have demonstrated that the density and the amount of blowing agent have a significant impact on the cellular structure of the foam. In turn, this structure determines the thermal conductivity of the foams. Using these thermal conductivity data, the authors worked out a qualitative model to determine the density gradients. Despite of quite self-consistency of the work in terms of accurate sample preparation, as well as the techniques used and data interpretation, the work would further gain in scientific soundness, if the results were added by molecular level studies, undermining the observed mesoscopic behavior of the samples.
In general, the work is of quiet high quality, provides new insights into the subject studied, has high application relevance, relies on an extensive referencing, so it is suitable for publication in its present form. The only recommendation is to chek the text for misprints, for instance:
Line 76 - 4.9​ m (seems incomplete)
Line 92 release - releases
Line 308 than - that
Line 20 acronym LDPE is not defined
etc.

Author Response

 Dear Reviewer, Polymers-MDPI

 ID# Polymers-1901227

On behalf of our research team, I thank you for considering our work.

In communication to our previously submitted manuscript with Polymers (21st of August, 2022) and the Reviewer # 3 (29 Aug 2022 18:03:38) gave the following comments.

I am attaching the reply for your comments.

Thank you very much

Saravana

Round 2

Reviewer 2 Report

Can be accepted now.